

# Assessing the potential of complex artificial neural networks for modelling small-scale soil erosion by water

Nils Barthel[1], Simone Ott[1], Benjamin Burkhard[1], and Bastian Steinhoff-Knopp[2]

[1]Leibniz University Hannover, Institute of Earth System Sciences, Physical Geography and Landscape Ecology Section, Schneiderberg 50, 30167, Hannover, Germany
[2]Thünen-Institute, Coordination Unit Climate Soil Biodiversity, Bundesallee 49, 38116, Braunschweig, Germany

**Correspondence:** Nils Barthel (barthel@phygeo.uni-hannover.de)

**Abstract.** Accurately modelling soil erosion by water is essential for developing effective mitigation strategies and preventing on- and off-site damages in agricultural areas. So far, complex artificial neural networks have rarely been applied in small-scale soil erosion modelling, and their potential still remains unclear. This study compares the performance of different neural network architectures for modelling soil erosion by water at a small spatial scale in agricultural cropland. The analysis is

based on erosion rate data at a 5 m × 5 m resolution, derived from a 20-year monitoring programme, and covers 458 hectares of cropland across six investigation areas in northern Germany. Nineteen predictor variables related to topography, climate, management and soil properties were selected as inputs to assess their interrelationships with observed erosion patterns and to predict continuous soil erosion rates. A single-layer neural network (SNN), a deep neural network (DNN), and a convolutional neural network (CNN) were applied and evaluated against a random forest (RF) model used as a benchmark. All machine

learning models have successfully captured spatial patterns of soil erosion, with the CNN consistently outperforming the others across all evaluation metrics. The CNN achieves the lowest root mean squared error (RMSE: 1.05) and mean absolute error (MAE: 0.41), outperforming the RF (RMSE: 1.31, MAE: 0.58) and the SNN (RMSE: 1.48, MAE: 0.63), while the DNN performs similarly to the CNN with a slightly higher RMSE (1.1) and MAE (0.45). The CNN notably outperforms the other three approaches when evaluating their capability to accurately predict soil erosion within given classes, achieving a weighted

mean F1 score of 0.7. A permutation importance analysis identified the digital elevation model as the most influential predictor variable across all models, contributing between 15 % and 18.3 %, while USLE C and R factors also had significant importance. Overall, these findings highlight the potential of complex neural networks for predicting spatially explicit rates of soil erosion by water.

## 1 Introduction

Soil erosion by water causes the loss of topsoil, along with the displacement of organic carbon and nutrients, degrading agricultural land and contributing to off-site damages such as water eutrophication and road blockages (Issaka and Ashraf, 2017). These processes have significant and long-lasting environmental and economic consequences in affected areas worldwide. A combination of anthropogenic, climatic, and topographic factors unique to each region defines the extent and distribution of soil erosion by water. Accurately modelling these interactions at a small scale can help identify key drivers and affected areas.



The resulting insights can support the development of targeted strategies to mitigate further soil degradation (Borrelli et al., 2018; Igwe et al., 2017).

Various approaches have been applied to predict soil erosion by water across different spatial scales, ranging from global (Guerra et al., 2020), continental (Panagos et al., 2021), national (Plambeck, 2020), to small field plots (Anache et al., 2018). The respective modelling approaches vary in data requirements, input variables, underlying principles, and complexity. Model

outputs are typically classified into erosion severity, susceptibility categories or expressed as continuous predictions of soil erosion rates over space and time. Depending on the intended output and available data, a variety of model types have been developed, including process-based, empirical, and, more recently, artificial intelligence (AI) models (Borrelli et al., 2021).

Examples of physical process-based models are WEPP (Nearing et al., 1989; Pieri et al., 2007), Erosion 3D (Schmidt et al., 1999), and EuroSEM (Morgan et al., 1998). Empirical models include the Universal Soil Loss Equation (USLE; Wischmeier

and Smith 1978), its revised version (RUSLE; Renard 1997), and further adaptations (Borrelli et al., 2021). One reason for the popularity of the USLE and its variations is their relatively low complexity, making these models time- and cost-efficient. However, the simplicity of traditional approaches like the USLE comes with limitations, as the model considers only a limited number of variables and often exhibits comparably low accuracy (Alewell et al., 2019; Avand et al., 2023; Kumar et al., 2022).

AI-supported modelling approaches have increasingly been used to overcome these limitations and to develop new methods

for identifying areas at risk of soil erosion by water. This includes the application of machine learning techniques such as random forest (RF; Garosi et al. 2019; Ghosh and Maiti 2021; Jaafari et al. 2022), with results indicating that RF outperforms methods such as support vector machines (SVMs) and generalized additive models (GAMs). While these studies highlight the potential of RF models, their main focus is on predicting and classifying gully erosion susceptibility.

Another promising AI approach for soil erosion modelling is the application of artificial neural networks (ANNs). However,

existing studies in this field have often employed simple neural network architectures, typically with only a single hidden layer. These single-layer networks have not only been used to model soil erosion susceptibility (De la Rosa et al., 1999) but also to predict quantitative soil erosion rates at the plot scale (Licznar and Nearing, 2003). These early applications demonstrate the potential of ANNs, but single-layer networks may not fully utilize the capacity of more complex architectures to capture non-linear relationships between influencing variables and the spatial distribution and magnitude of soil erosion (Avand et al.,

2023).

More complex, multi-layered ANNs have rarely been applied to predict soil erosion rates quantitatively. Instead, they have been mostly used to model a classified erosion susceptibility (Golkarian et al., 2023; Khosravi et al., 2023; Sarkar and Mishra, 2018) or have been restricted to gully erosion (Ghorbanzadeh et al., 2020; Saha et al., 2021). In cases where ANNs have been applied to quantify continuous soil erosion rates, studies often rely on limited datasets, such as those collected using erosion

pins over just one year (Gholami et al., 2021; Sahour et al., 2021).

While machine learning methods have shown promise in modelling soil erosion, their application to predict continuous erosion rates at fine spatial scales remains limited, particularly when using neural networks. Currently, most studies rely on input data with spatial resolutions between 5 and 300 metres, with higher-resolution datasets often restricted to small areas, such as individual plots (Borrelli et al., 2021; Parsons, 2019). This lack of research combining neural networks with high-





resolution data is partly due to the limited availability of long-term monitoring schemes at the landscape scale, which provides spatially explicit erosion data as ground truth for model training and validation (Batista et al., 2025).

To address this gap, this study utilized existing long-term soil erosion monitoring data which spans over two decades and was collected across seven study areas in northern Germany (Steinhoff-Knopp and Burkhard, 2018). This monitoring dataset was combined with a range of high-spatial-resolution predictor variables as input for different machine learning approaches. Specifically, three neural networks of increasing complexity and a RF model were systematically compared, evaluating their predictive accuracy on determining continuous soil erosion rates.

The goal of this study was to contribute to a more comprehensive understanding of the strengths and limitations of different AI-based approaches for soil erosion modelling by aiming to answer the following research questions:

1. Which machine learning approach exhibits the highest predictive performance in estimating water-induced soil erosion rates?

2. What are the challenges and limitations in predicting soil erosion by water at a small spatial scale using machine learning models?

3. Which variables are most important for predicting soil erosion by water at field to landscape scale across different machine learning models?

## 2 Methods

### 2.1 Study area

The models are based on data from seven distinct study areas across Lower Saxony, Germany and are located between 53.1° N and 51.9° N latitude and 8.3° E and 10.5° E longitude. Five of these regions are clustered in the southeastern part of Lower Saxony, and named *Leine-Innerstebergland*, while the other two are located in the Northeast and Southwest (Fig. 1). The cultivated cropland within all the regions covers 458 hectares, is prone to erosion, and represents different soil types, relief characteristics, and management conditions (Capelle and Lüders, 1985; Capelle, 1990).

A large part of the soils – predominantly Luvisols and Cambisols – has a high silt content due to loess or sand-loess deposits, increasing the erodibility of the topsoil (Steinhoff-Knopp and Burkhard, 2018). The mean slope across all study areas is 4.01°, with the southern regions having a steeper relief (4.81°) compared to the northern (2.35°) and the western region (3.68°). The primary crop is winter wheat, followed by winter barley, rapeseed, sugar beet, maize, and potatoes. All farmers within the monitored areas practice conventional farming and implement various soil conservation practices, including reduced tillage, contour-parallel tillage, cover crops, grassed tramlines, and drainage systems.





**Figure 1.** Overview of study areas in Lower Saxony and mapped soil erosion rates in (a) Küingdorf, (b) Barum, (c) Lamspringe, (d) Klein Ilde, (e) Nette, (f) Adenstedt, and (g) Brüggen.





## 2.2 Data collection

The soil erosion dataset used in this study was derived from a long-term monitoring programme funded by the Lower Saxony
State Authority for Mining, Energy and Geology (LBEG), covering the years 2000 to 2020. Exceptions include the investigation
areas Adenstedt, where monitoring began in 2002, and Klein Ilde, where monitoring stopped in 2015. The methodology of the
field surveys is based on the recommendations by Rohr et al. (1990) and the instructions by DVWK (1996) and Botschek et al.
(2021). To implement a more efficient workflow, the mobile mapping application *EroPad* was used since 2010 (see Steinhoff
et al. 2013). The field surveys were carried out each year after the snowmelt (usually in February or March) and usually within
95 one week after each erosive rainfall event ($> 12.7\,mm\,h^{-1}$; see DIN 19708:2022-08) throughout the summer months.

The field surveys aimed to collect data on the spatial distribution and extent of three main types of soil erosion by water: linear
(i.e. rill) erosion, sheet erosion, and sheet-to-linear erosion. The losses by linear erosion features were surveyed by measuring
their volume and extrapolated to an 8 m buffer around their occurrence to account for the dispersion of soil loss due to tillage
practices by farmers (Steinhoff-Knopp and Burkhard, 2018). For the modelling process, soil erosion rates ($t\,ha^{-1}yr^{-1}$) were
100 calculated by overlaying all surveyed erosion features in a grid resolution of 5 m × 5 m (Fig. 1).

Nineteen predictor variables (i.e. features or covariates) potentially influencing soil erosion were selected for the modelling
process (Table 1). These variables include factors related to topography, climate, soil properties, and agricultural land man-
agement. The digital elevation model (DEM) and all derived topographic variables were processed at their native resolution of
5 m × 5 m. The USLE R-factor (erosivity of rainfall), originally at a grid resolution of 1 km × 1 km, was resampled to match
the DEM-based variables. The USLE K-factor (erodibility of topsoil), derived from a soil map at a scale of 1:50 000, and the
USLE C-factor (crop cover and management factor) were rasterized to the 5 m × 5 m grid. The C-factor was determined for
each field using agricultural land management data collected alongside the field surveys.

Several of the variables, especially the topographic variables, were derived from each other or capture related aspects,
which could lead to a strong correlation between them (Avand et al., 2023; Jaafari et al., 2022). Therefore, the degree of
the linear relationship between variables was assessed by using the Pearson coefficient (r) and the variance inflation factor
(VIF). The Pearson coefficient is used to gain information on the pairwise linear correlation between the variables, with ±0.7
being considered a threshold for strong correlation (Schober et al., 2018). The VIF is used to assess multicollinearity across
all variables, evaluating whether variables with high pairwise correlations can still provide valuable information for the final
predictions (Ebrahimi-Khusfi et al., 2021; O'brien, 2007). Variables with a VIF below 5 are considered to have a low level of
multicollinearity and are used for the modelling (Daoud, 2017).

## 2.3 Machine Learning Models

This section provides a brief introduction to the four distinct machine learning models employed in this study. The model
implementations and hyperparameter selection process are described in Sect. 2.5.



| Variable | Description | Unit |
|---|---|---|
| DEM | Digital elevation model of the terrain | m |
| Slope | Steepness of the terrain surface | degree (°) |
| Slope length | Horizontal length of a slope | m |
| Aspect 360 | Direction a slope faces [0 - 360°] | degree (°) |
| Aspect 180 | Slope direction normalized to 180° [0 - 180°] | degree (°) |
| Plan curvature | Horizontal curvature affecting water flow | - |
| Profile curvature | Curvature along the slope direction | - |
| Flow line curvature | Curvature of flow lines across the terrain | - |
| Topographic position index (TPI) | Position relative to surrounding terrain | - |
| Flow accumulation | Total accumulated runoff | - |
| Wetness index | Index based on a modified catchment area calculation | - |
| Divergence-convergence index (DCI) | Index of water flow divergence or convergence, based on a 3×3 cell area | - |
| DCI 10 | Index based on a 10×10 cell area | - |
| Machining direction (MD) | Orientation of the machining direction (tramlines) [0 - 180°] | degree (°) |
| MD vs. aspect | Angle between MD and Aspect 180 [0 - 90°] | degree (°) |
| R factor | USLE rainfall erosivity factor | $\mathrm{MJ\,mm\,(h\,ha\,yr)}^{-1}$ |
| K factor | USLE soil erodibility factor | $\mathrm{t\,h\,(MJ\,mm)}^{-1}$ |
| LS factor | USLE topographic factor (slope length and steepness) | - |
| C factor | USLE crop and management factor | - |

**Table 1.** Variables used in the soil erosion modelling process.

### 2.3.1 Random forest

A random forest (RF) model combines multiple decision trees to create an ensemble model to improve prediction accuracy (Breiman, 2001). A decision tree splits data based on feature values across multiple levels of nodes, with each branch representing different decision paths leading to predictions (Kingsford and Salzberg, 2008). Each tree is trained on a random subset of the data, and, in the case of regression tasks, as applied in this study, the final prediction is obtained by averaging the outputs of all trees.

Although mostly limited to classification outputs, RF has shown promise in previous studies on soil erosion modelling (Garosi et al., 2019; Ghosh and Maiti, 2021; Jaafari et al., 2022). Therefore, RF serves as a benchmark for evaluating the performance of neural networks against other machine learning models. The RF model used in this study included 500 decision trees, a maximum tree depth of 10, a minimum of two samples required to split an internal node and a minimum of one sample per leaf.





### 2.3.2 Single hidden layer neural network

Artificial neural networks consist of an input layer, an output layer, and one or more so-called hidden layers of neurons (or nodes) in between (Rumelhart et al., 1986). Each neuron in these layers is interconnected through weights. During training, these weights are updated through an optimization process involving multiple iterations, a loss function, and most commonly, backpropagation. The weighted sum of the inputs is passed through an activation function, which applies a non-linear transformation to determine the neuron's output, which is then passed on to the subsequent layer. In a fully connected (or dense) layer, each neuron is connected to every neuron in the subsequent layer through individual weights.

Three different types of neural networks were compared in this study, each with a distinct and increasingly complex architecture. The first type is a neural network with a single dense layer (SNN) of 64 neurons between the input and the output layer (see Wythoff 1993). The Rectified Linear Unit (ReLU) activation function was applied in both the hidden and output layers in this and the following two variations of neural network models, to introduce non-linear relationships and constrain predictions to positive values (see Krizhevsky et al. 2017; Nair and Hinton 2010). Additionally, $L_2$ regularisation, also known as Ridge regularisation, was employed across all neural networks to reduce overfitting (see Hoerl and Kennard 1970; Ng 2004). The Adam optimizer was used to determine the learning rate during training (see Kingma and Ba 2014).

### 2.3.3 Deep neural network

The interconnected neurons and the corresponding weights enable neural networks to capture complex non-linear relationships. Increasing the number of hidden layers enhances this capability while increasing training time and the likelihood of overfitting the model to the training data. Such a neural network with multiple hidden layers is commonly known as a deep neural network (DNN) or a deep learning model (LeCun et al., 2015). In this case, the DNN was modified to have three hidden layers, with 64 neurons in the first, 128 neurons in the second, and 256 neurons in the third hidden layer.

### 2.3.4 Convolutional neural network

Convolutional neural networks (CNNs) were specifically designed to capture spatial relationships in grid-like structures, such as images (LeCun et al., 2015). This is achieved through convolutional layers, which use filters, also referred to as kernels. Filters are small matrices (e.g. $5 \times 5$ raster cells) that slide across the grid data, applying the same set of weights across the entire grid. By doing so, CNNs are an effective and established approach to detect patterns across grid data (Krizhevsky et al., 2017; LeCun et al., 2015). In this study's CNN, three convolutional layers were used, each employing 256 kernels. Two dense layers with 128 neurons each were added following the convolutional layers. The dense layers utilize the learned spatial patterns from the convolutional layers to produce the final predicted output.

### 2.4 Validation

The results were compared using different validation metrics. To determine the overall differences between the modelled and the mapped soil erosion rate, the root mean squared error (RMSE) and the mean absolute error (MAE) were used. The





F1 score, which is the harmonic mean of precision and recall, was used to evaluate the ability of the models to predict within different ranges of soil erosion severity (Chinchor, 1992). For this, the continuous predicted output was categorized into six discrete classes: no erosion ($0\,t\,ha^{-1}yr^{-1}$), very low erosion ($> 0$ to $< 0.25\,t\,ha^{-1}yr^{-1}$), low erosion ($0.25$ to $< 1\,t\,ha^{-1}yr^{-1}$), medium erosion ($1$ to $< 2\,t\,ha^{-1}yr^{-1}$), high erosion ($2$ to $< 5\,t\,ha^{-1}yr^{-1}$), and very high erosion ($\geq 5\,t\,ha^{-1}yr^{-1}$). A weighted

165 average F1 score was calculated to account for the imbalanced distribution of the mapped soil erosion dataset across the classes, with the majority (81.2 %) of the data falling below $1\,t\,ha^{-1}yr^{-1}$. The resulting F1 score ranges from 0 to 1, with values closer to 1 indicating a better overall alignment between the prediction and the different classes (Taha and Hanbury, 2015).

 The importance of the predictor variables was evaluated through permutation importance, which reflects the influence of each variable on the final model output. This is done by randomly permuting one variable at a time and measuring the resulting

170 increase in the mean squared error. A larger increase indicates that the variable has a greater impact on model accuracy. The results are normalized and expressed as percentages (Altmann et al., 2010).

### 2.5 Model implementation

The predictive performance of each model is highly influenced by the hyperparameters, which control the architecture and learning behaviour. The most important hyperparameters (e.g. number of trees or neurons) and their optimal values were de-

175 scribed along with the respective models above (Sect. 2.3). To determine the optimal value for each hyperparameter, grid search was used, iterating over a predefined range of parameter settings to identify the combination that yields the best performance (Yu and Zhu, 2020).

 To improve the reliability of the model results, five-fold cross-validation was used. In each fold, the model was trained on 80 % of the data and evaluated on the remaining 20 %, which was not used for training. The validation metrics were then averaged

180 across the five runs. All models were built using the scikit-learn (Pedregosa et al., 2011) and TensorFlow (Abadi et al., 2015) Python packages.

### 3 Results

### 3.1 Pairwise correlation and multicollinearity analysis

To assess relationships between predictor variables and identify potential redundancy due to multicollinearity, the Pearson

185 correlation coefficient (r) and the variance inflation factor (VIF) were calculated. Five pairs of variables (see Table 1) had a strong linear correlation with Pearson coefficient close to or exceeding $\pm 0.7$: aspect 360 and aspect 180 ($r = 0.69$), slope and wetness index ($r = -0.69$), DEM and R factor ($r = -0.75$), topographic position index and divergence-convergence index 10 ($r = 0.82$), slope length and LS factor ($r = 0.82$). However, all variables had a VIF $< 5$, indicating that there is not a high level of multicollinearity between the variables. Consequently, no variables were removed for the modelling process.





## 3.2 Model performance

The results show that the CNN model outperforms all three other models across all validation metrics (Fig. 2 and Table 2). When comparing the continuous modelled output to the mapped soil erosion rates, the CNN achieves the lowest error, with an RMSE of 1.05 and an MAE of 0.41, which are notably lower than those of the RF (RMSE: 1.31, MAE: 0.58) and SNN (RMSE: 1.48, MAE: 0.63) models. Only the DNN performed similarly, with a slightly higher RMSE of 1.10 and MAE of 0.45.

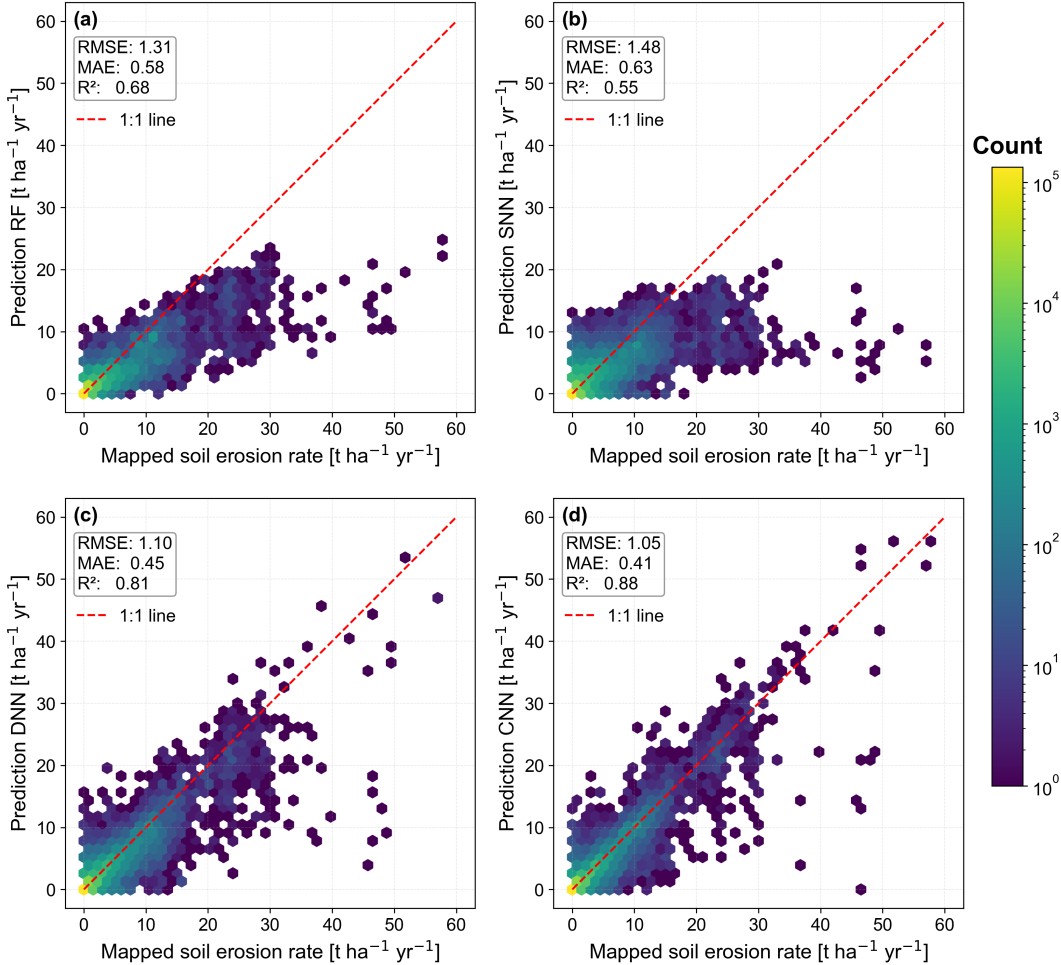

**Figure 2.** Plots comparing the model predictions (y-axis) against the true mapped values (x-axis) for (a) random forest (RF), (b) single-layer neural network (SNN), (c) deep neural network (DNN), and (d) convolutional neural network (CNN).

The difference between the model predictions and the mapped soil erosion rates (i.e., "true values") is visualised in Fig. 2, where predicted values are plotted on the y-axis against the observed true values on the x-axis, illustrating their linear correlation. It becomes clear that the RF (Fig. 2a) and SNN (Fig. 2b) models tended to underestimate areas with high mapped





soil erosion rates. In contrast, the DNN (Fig. 2c) and CNN (Fig. 2d) provided a closer fit to the ground truth data, including areas with higher erosion rates. This is supported by the $R^2$ values of 0.81 for the DNN and 0.88 for the CNN, indicating that

both models captured a large portion of the variability in the mapped values. While the CNN model produced the most reliable results when predicting continuous soil erosion rates, it still occasionally exhibited large prediction errors with a varying spatial distribution both between and within the seven study areas.

**Figure 3.** Difference between soil erosion rates predicted by the convolutional neural network (CNN) and the mapped values in (a) Küingdorf, (b) Barum, (c) Lamspringe, (d) Klein Ilde, (e) Nette, (f) Adenstedt, and (g) Brüggen, where positive values indicate overestimation of the model and negative values indicate underestimation.





The spatial distribution of these differences between the predicted continuous results of the CNN model and the mapped soil erosion is shown in Fig. 3. Especially in the study areas Barum, Lamspringe, and Klein Ilde, there are notable differences be-

205 tween the modelled and the mapped soil erosion rates. These large over- and underestimations of soil erosion often occurred in close spatial proximity. In contrast, in the areas of Küingdorf, Nette, Adenstedt, and Brüggen, the difference between modelled and mapped soil erosion was mostly below 5 $tha^{-1}yr^{-1}$, with no large over- and underestimations occurring simultaneously in close proximity to each other.

By comparing the log-transformed cumulative distributions of mapped soil erosion rates and the model predictions, distinc-

210 tions in the capabilities of the different models become even more evident (Fig. 4). The main differences between the models are visible in areas with very low mapped soil erosion rates ($< 0.25\,tha^{-1}yr^{-1}$). Here, the CNN aligned most closely with the mapped soil erosion rates, followed by the RF model. In contrast, the DNN and SNN models seemed unable to accurately predict very low soil erosion rates. Instead, DNN and SNN frequently predicted a soil erosion rate of 0 $tha^{-1}yr^{-1}$, which rarely occurs in the ground truth dataset (see Table 2).

This distinction in model performance for low soil erosion rates is further highlighted by the F1 scores, which indicate how well the continuous predictions of each model align with the defined classes of soil erosion severity. The CNN achieved a higher F1 score in every class compared to the other three models, indicating consistently better performance across all classes. This is reflected in the weighted average F1 score, where CNN had the highest score (0.70), followed by RF (0.49), DNN (0.25), and SNN (0.13).

**Table 2.** F1 score evaluating the performance of the continuous model predictions (in $tha^{-1}yr^{-1}$) within the defined soil-erosion classes, comparing random forest (RF), single-layer neural network (SNN), deep neural network (DNN) and convolutional neural network (CNN) models. The weighted average F1 scores were computed using the number of mapped grid-cells (n cells) within each class.

| | | F1 score | | | |
|---|---|---|---|---|---|
| Soil erosion rate | n cells | RF | SNN | DNN | CNN |
| 0 | 655 | 0.00 | 0.01 | 0.01 | 0.22 |
| $< 0.25$ | 113 105 | 0.54 | 0.04 | 0.09 | 0.83 |
| $0.25 - < 0.5$ | 22 815 | 0.27 | 0.07 | 0.17 | 0.41 |
| $0.5 - < 1.0$ | 13 912 | 0.24 | 0.12 | 0.22 | 0.31 |
| $1.0 - < 2.5$ | 17 849 | 0.37 | 0.29 | 0.45 | 0.55 |
| $2.5 - < 5.0$ | 10 032 | 0.40 | 0.37 | 0.55 | 0.65 |
| $\geq 5.0$ | 7 073 | 0.56 | 0.56 | 0.75 | 0.81 |
| **Weighted average** | | 0.46 | 0.11 | 0.20 | 0.70 |





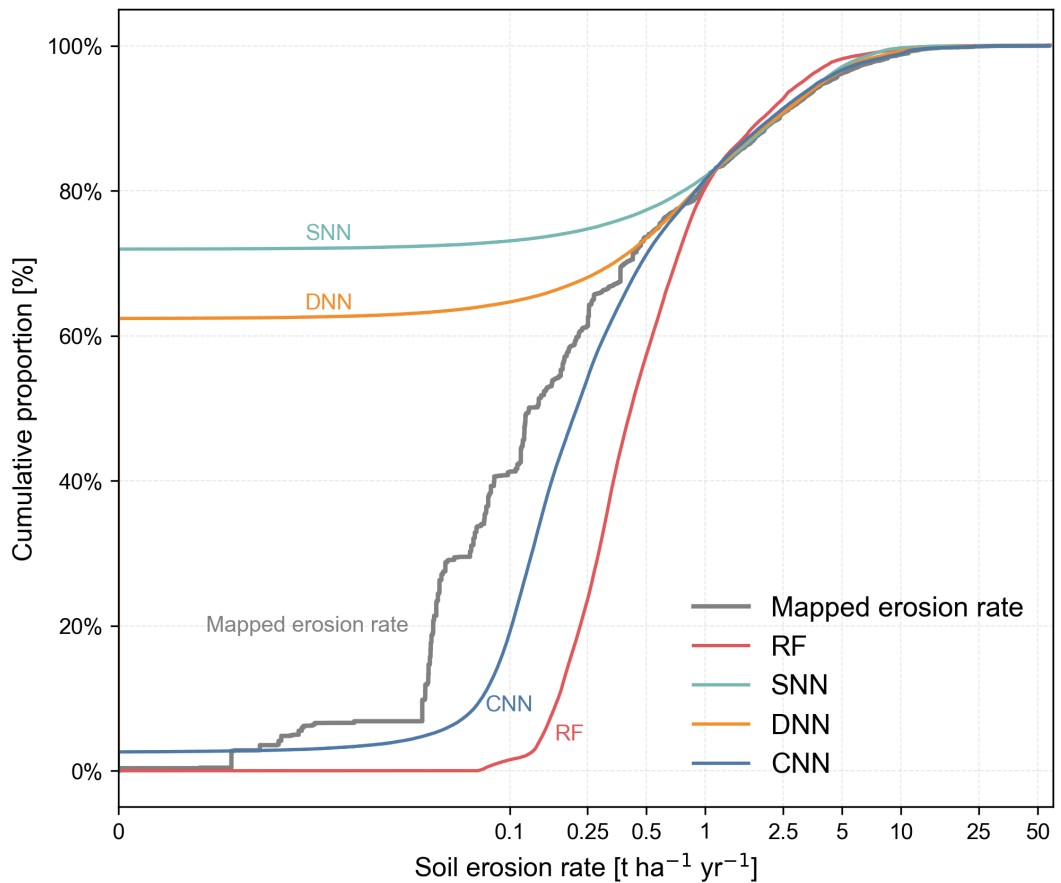

**Figure 4.** Empirical cumulative distribution functions (ECDFs) of soil erosion rates for observed mapped soil loss, random forest (RF), single-layer neural network (SNN), deep neural network (DNN) and convolutional neural Network (CNN).

## 3.3 Variable importance

The results of the permutation importance analysis are visualised in Fig. 5 and compared across all models, from the CNN model with the highest predictive performance on the left to the SNN model with the lowest performance on the right. The predictor variables are ranked from top to bottom based on their importance to the CNN model.

The variable with the highest importance was the DEM in all four models. Its importance ranges from 15 % in the DNN to 18.3 % in the CNN. All models also strongly relied on the C and R factors, originating from the empirical erosion model USLE. One distinct difference between the neural network-based models and the RF model was the importance of the variables machining direction vs. aspect and the USLE LS factor. While the three neural network models assigned very low importance or did not rely on these variables at all, they had a moderate importance of 7.5 % and 6.3 %, respectively, in the RF model.





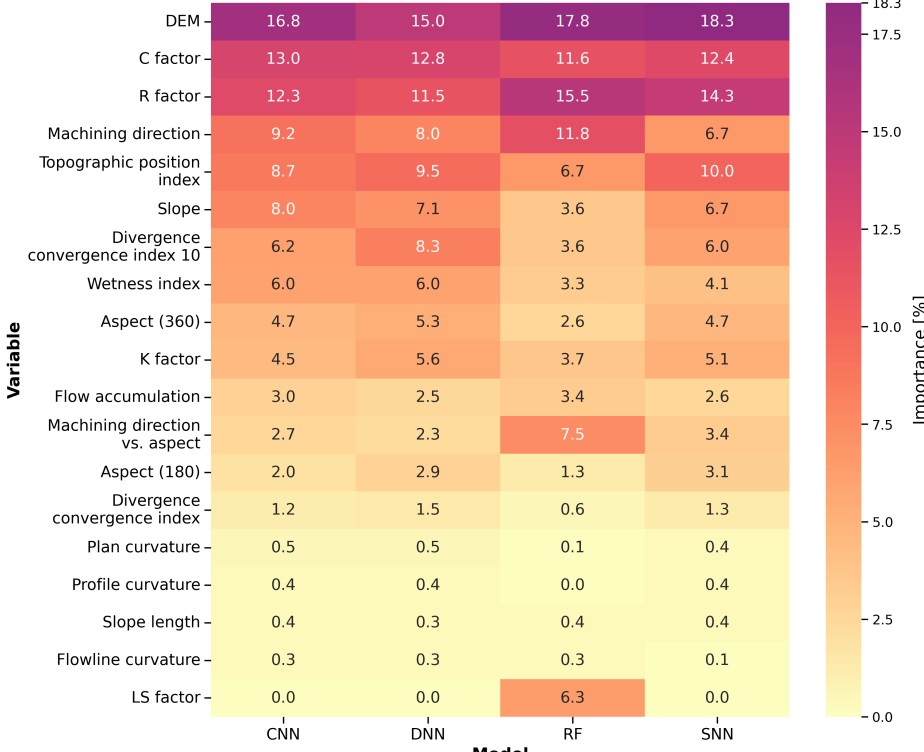

**Figure 5.** Heatmap showing the permutation importance of predictor variables across four modelling approaches: convolutional neural network (CNN), deep neural network (DNN), random forest (RF), and single-layer neural network (SNN). Models are ordered from left (highest predictive performance) to right (lowest performance), and variables are ranked by their importance in the CNN model.

Variables with very low importance ($< 2$ %) across all models were plan curvature, profile curvature, slope length, and flowline curvature.

## 4 Discussion

### 4.1 Comparison of Model Performance

The results concur with previous studies, such as Sarkar and Mishra (2018), indicating that increasing complexity through multiple hidden layers in a DNN can improve predictive performance compared to a SNN. Large neural networks have the advantage of being able to capture the non-linear interrelationships between variables and the predicted output using an extensive set of weights. This allows to model these relationships in greater detail, which, as shown in the results, can improve the accuracy of the predictions. Similar to the findings by Golkarian et al. (2023), our results showed that the predictive performance



increases further through the addition of convolutional layers in the CNN, which utilizes filters and shared weights to capture spatial patterns.

Both RMSE and MAE values indicate that the CNN and DNN had higher predictive performance than the RF model. As shown in Fig. 4 and Table 2, the RF model performed second best after the CNN when considering prediction performance across different classes of soil erosion severity. However, the RF model failed to capture very high loss rates exceeding 20 $t\,ha^{-1}yr^{-1}$, resulting in low RMSE and MAE. While the DNN achieved RMSE and MAE values similar to those of the CNN, its weighted F1 score was low, which can be explained by the imbalance in the mapped erosion data. A large proportion

(81.2 %) of the mapped data is below the threshold of 1 $t\,ha^{-1}yr^{-1}$ and consequently this value range had a large impact on the weighted average F1 score. Thus, by underestimating these low soil erosion rates, the DNN achieved a lower weighted average F1 score compared to the CNN and RF models. Although mostly attributed to accurate predictions of lower soil erosion rates, the weighted average F1 score of 0.49 suggests that the RF model remains at least somewhat effective when predictions are evaluated as classified outputs, which is consistent with previous studies such as Garosi et al. (2019) and Sahour et al. (2021).

Overall, this study reveals that the CNN outperformed all other tested models across all validation metrics. Although the DNN achieved similar RMSE and MAE values, the CNN demonstrated a higher accuracy in capturing lower soil erosion rates, while the SNN failed by predicting large areas with no soil erosion at all. However, it is important to note that the DNN, and particularly the SNN and RF models, require significantly fewer computational resources than the CNN, which can be beneficial when trained on big datasets or when the model is applied to predict soil erosion for larger areas. Therefore, depending

on the specific use case, the potential trade-off between improved predictive performance and the increased complexity and computational demands of large neural networks such as the CNN should be carefully considered.

The permutation importance analysis highlights the combined relevance of topographic, climatic, and anthropogenic (i.e., agricultural land management) variables in influencing soil erosion. The variation in importance assigned to the LS factor by the RF model compared to the other models can be explained by the fundamental differences in how tree-based and neural

network-based models generate their predictions. On the other hand, the consistently high permutation importance (> 5 %) of the DEM, C and R factor, machining direction, and the topographic position index across all models underscore their influence on soil erosion by water processes and the resulting soil erosion rates and patterns. These findings are consistent with studies such as Bag et al. (2022) and Wang et al. (2020), although differences in study areas conditions, research objectives, and variable selection make direct comparisons across studies challenging.

## 4.2  Limitations and Future Research

Typical challenges and limitations in soil erosion modelling at the landscape scale include the high range of input variables from different domains for parametrization, leading to simplification in variable estimation and mismatches in the scale of input and output data, the application of models outside of the validation range and inherent restrictions of the used models. This study addressed several of these limitations by utilizing a dataset based on a long-term soil erosion monitoring programme

across multiple study areas, capturing long-term patterns of soil erosion by water within these regions. Additionally, input variables with relatively high spatial resolution were incorporated into machine learning models capable of capturing non-





linear relationships at varying levels of complexity. Despite these approaches, several challenges and limitations remain. These challenges and limitations should be considered when interpreting the results and highlight key areas for future research to enhance small-scale soil erosion predictions.

The results of this study show notable over- and underestimations occurring in close spatial proximity (Fig. 3). This is likely due to the inability of the models to effectively capture the differing scales of soil erosion within small areas. Although the spatial resolution of the input data is relatively high, increasing it further to a grid resolution of, for example, a resolution of 1 m × 1 m could improve the ability of the models to capture more fine-scale variations in soil erosion patterns. Nonetheless, certain small-scale variables remain difficult to incorporate and are highly variable in time and space, including the annual

placement of tramlines and temporal variability in crop or ground cover throughout the year.

    The results indicate furthermore that prediction accuracy improves with increasing model complexity (Fig. 2), although the difference between CNN and DNN was relatively small. However, the DNN model was particularly unable to predict very low soil erosion rates and often incorrectly predicted no soil erosion at all (Table 2). Whether this generally justifies the use of a more complex CNN architecture in soil erosion modelling must be carefully evaluated based on the aim and context of

the respective study. Future research could explore this further by testing and comparing different multi-layer neural networks, including more novel architectures such as transformers (e.g. Liu et al. 2024).

    In particular, large neural networks require large datasets and regularisation to prevent overfitting and improve the generalizability of their predictions. Whether the dataset size and regularisation strength used in this study were sufficient to allow for reliable extrapolation of the modelled output to other agricultural areas remains to be tested. Future work should assess

how well these models perform when applied to a larger area and whether additional data or adjustments in complexity and regularisation are needed to ensure good prediction results, e.g. for agricultural areas in the entire state of Lower Saxony. In addition, it would be valuable to expand the monitoring dataset used for training, although newly monitored areas will take time to accurately reflect long-term patterns of soil erosion by water.

## 5   Conclusions

By comparing different artificial neural network architectures and using a random forest model as a benchmark, this study highlights the potential of complex neural networks for modelling soil erosion at a small spatial scale. The convolutional neural network (CNN) outperformed all other models, achieving a comparatively low RMSE of 1.05 and the highest F1 score of 0.70. Although the deep neural network (DNN) without convolutional layers achieved a similarly low RMSE of 1.10, the F1 scores indicate that it was unable to accurately predict low soil erosion rates. In contrast, the random forest model performed

better in this regard while also being more computationally efficient. All tested models were highly dependent on the DEM as the variable with the highest permutation importance, followed by the USLE C and R factors.

    The results suggest that simpler neural networks may lead to lower predictive performances and that more advanced approaches could offer further improvements in soil erosion modelling. Nevertheless, less complex machine learning models are



still capable of predicting classified soil erosion rates. To build upon these findings, future research should focus on evaluating

the generalizability of these models and their scalability to larger areas.





*Code and data availability.* The complete dataset and Python scripts used for model training, prediction, and evaluation are available at:

https://doi.org/10.5281/zenodo.16032629 (Barthel, 2025).



## Appendix A: Pairwise correlation and multicollinearity analysis



**Figure A1.** Pairwise linear correlation analysis of all predictor variables used in this study based on the Pearson correlation coefficient.



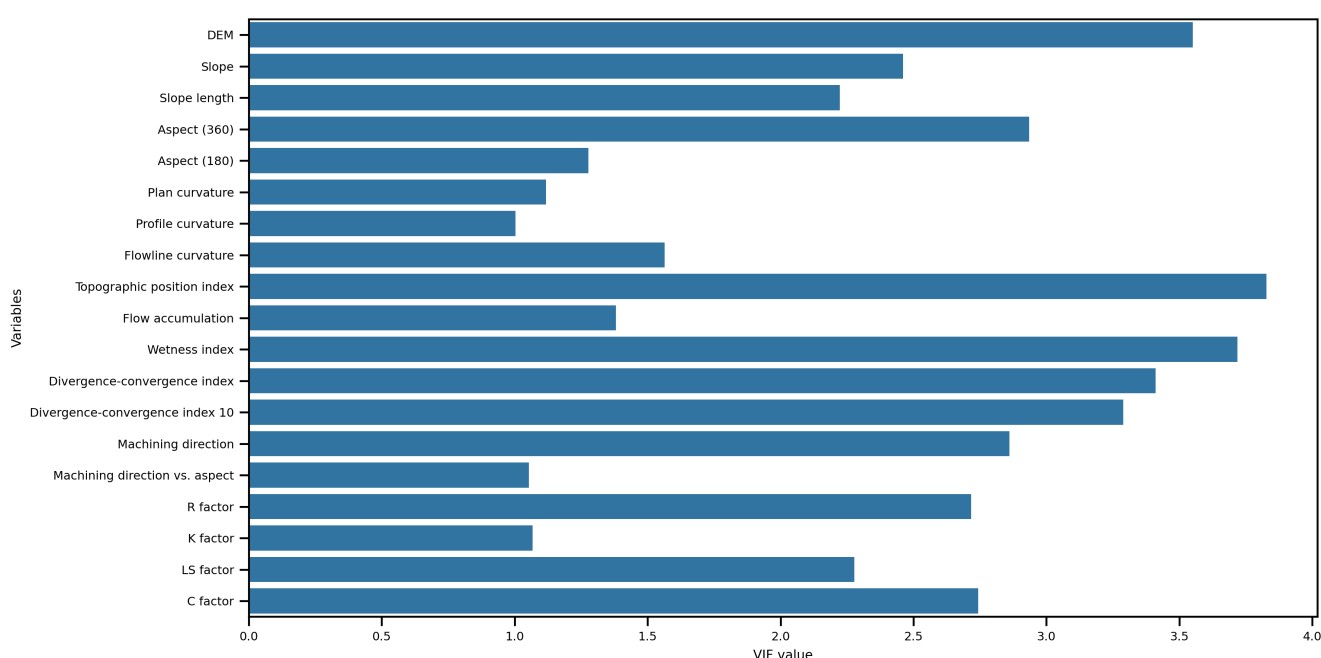

**Figure A2.** Variance inflation analysis of predictor variables highlights the level of multicollinearity, with higher VIF values indicating greater multicollinearity among predictors.



## Appendix B: Model performance



**Figure B1.** Difference between soil erosion rates predicted by the random forest (RF) and the mapped values in (a) Küingdorf, (b) Barum, (c) Lamspringe, (d) Klein Ilde, (e) Nette, (f) Adenstedt, and (g) Brüggen, where positive values indicate overestimation of the model and negative values indicate underestimation.



**Figure B2.** Difference between soil erosion rates predicted by the single-layer neural network (SNN) and the mapped values in (a) Küingdorf, (b) Barum, (c) Lamspringe, (d) Klein Ilde, (e) Nette, (f) Adenstedt, and (g) Brüggen, where positive values indicate overestimation of the model and negative values indicate underestimation.





**Figure B3.** Difference between soil erosion rates predicted by the deep neural network (DNN) and the mapped values in (a) Küingdorf, (b) Barum, (c) Lamspringe, (d) Klein Ilde, (e) Nette, (f) Adenstedt, and (g) Brüggen, where positive values indicate overestimation of the model and negative values indicate underestimation.



*Author contributions.* NB: conceptualization, methodology, investigation, visualization, writing (original draft), writing (review and editing). SO: conceptualization, data curation, writing (review and editing). BB: conceptualization, project administration, funding acquisition, writing (review and editing). BSK: conceptualization, data curation, visualization, writing (review and editing).

*Competing interests.* The contact author has declared that none of the authors has any competing interests.

*Acknowledgements.* The research and collection of the associated data were made possible through the continuous funding provided by the

Lower Saxony State Authority for Mining, Energy and Geology (LBEG). We would also like to acknowledge the many years of work by all those who have contributed to the creation of this long-term dataset through fieldwork since 2000. In particular, we thank Frank Beisiegel and Heiko van Wensen for their continuous monitoring efforts since the beginning of the programme. We thank Angie Faust for proofreading this work.



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
