# Peer review of "Assessing the potential of complex artificial neural networks for modelling small-scale soil erosion by water"

_EGUsphere, 2025_

## Author Comment (AC1)

Dear Reviewer,

We sincerely thank you for your detailed and constructive feedback. We have carefully considered each of your comments and will revise the manuscript accordingly. We are pleased that you recognize the value of our work and are confident that the revisions made in response to your suggestions will further strengthen the manuscript. Please find our detailed responses to each of your points below.

**Reviewer comments**

The authors emphasise in several places that their goal is an accurate soil erosion prediction. If this is their goal, they fail and will always fail simply because soil erosion and its drivers are random quantities (in a statistical sense). Random quantities, by definition, are never accurate. Hence, the authors require a more realistic goal.

The authors write that the USLE comes with comparable low accuracy. This is clearly wrong, although this claim can often be found in literature. The USLE comes with the best possible accuracy for a random quantity. It is the modeller who fails because they apply the USLE with poor data or poor knowledge of its usage. It may be that the data needed to obtain good results are unavailable, but this is not the fault of the USLE. The data have to be gathered by the modeller. If we want to progress, it is essential to be more precise in describing deficits. Machine learning will likely not improve the modellers and their available data.

**Response**

Thank you for bringing this to our attention. We agree that our statements can be improved through rephrasing and by providing additional detail. We do not claim to accurately predict individual soil erosion events, but rather to reproduce spatial patterns and average water erosion loss rates derived from a 20-year monitoring programme at the field-to-landscape scale. The long-term averages obtained in the monitoring are still random variables (in a statistical sense) but can be interpreted as empirical documented site characteristics defined by soil properties, topography, management, rainfall erosivity, etc..

Therefore, our modelling goal is to predict these patterns as accurately as possible relative to the available mapped data and to detect the underlying spatial relationships. Our focus is on describing relationships rather than achieving absolute accuracy. We acknowledge that our phrasing ("accurately modelling") may have implied an unrealistic level of accuracy.

Therefore, we will rephrase the relevant text to clarify that our contribution lies in improving the accuracy of soil erosion pattern modelling at the field-to-landscape scale.

We agree that our phrasing may have suggested that the USLE model inherently produces low accuracy, and we acknowledge that we need to be more precise when discussing the limitations of USLE applications. Indeed, applying the USLE with poor input data or in non-validated settings can lead to low accuracy, and it remains the modeller's responsibility to obtain high-quality data to achieve reliable results.

The same holds true for machine-learning approaches, which likewise do not release modellers from the need to collect reliable input data.

Our goal is to develop modelling approaches that operate at the field-to-landscape scale. The USLE was originally based on plot-scale measurements; its first versions were designed for single slopes to predict soil loss from sheet and small-rill erosion. Subsequent developments extended the USLE to the field and landscape scales, incorporating more complex slope geometries. Our approach builds directly on long-term field measurements and uses these data to train machine-learning models. In doing so, it is essential to better understand how relationships between variables influencing soil erosion patterns and loss rates can be represented within machine learning models for the estimation of soil loss (e.g., high-vs.

|                                                                                                                                                                                                                                                                                                                                                                                                                                                                                                                                                                                                                                                                                | <del></del>                                                                                                                                                                                                                                                                                                                                                                                                                                                                                                                                                                                                                                                                                                                                                                                                                                                                                                                                                                                                                                                                          |
|--------------------------------------------------------------------------------------------------------------------------------------------------------------------------------------------------------------------------------------------------------------------------------------------------------------------------------------------------------------------------------------------------------------------------------------------------------------------------------------------------------------------------------------------------------------------------------------------------------------------------------------------------------------------------------|--------------------------------------------------------------------------------------------------------------------------------------------------------------------------------------------------------------------------------------------------------------------------------------------------------------------------------------------------------------------------------------------------------------------------------------------------------------------------------------------------------------------------------------------------------------------------------------------------------------------------------------------------------------------------------------------------------------------------------------------------------------------------------------------------------------------------------------------------------------------------------------------------------------------------------------------------------------------------------------------------------------------------------------------------------------------------------------|
| The authors claim that Al-supported modelling approaches are increasingly applied to overcome the limitations of the USLE (limited number of variables, low accuracy). This is wrong. I am not aware of any publication that used an unlimited number of variables. I am unaware of any application of an Al modelling approach independent of its developer, as is the case with the application of the USLE. We do not know whether Al models will perform better when widely applied because these models are unavailable. Hence, a comparison with the USLE is presently impossible, and it is wrong to write that such comparisons exist. This is still a long way to go. | low-complexity, tree-based vs. neural-network structures) and to evaluate the advantages and disadvantages of different machine-learning approaches. We agree that our aims should be expressed more clearly and will revise the manuscript accordingly.  We apologize for our very short argumentation here and will rephrase the relevant paragraph (lines 33 – 38). We are also not aware of any publication claiming to substitute the USLE by a machine learning model. Most AI (or to be more precise ML) driven soil erosion approaches are designed to map erosion processes not modelled by the USLE or process-based models (we also refer to WEPP, EROSION 3D and EuroSEM), such as gully erosion.  In our study, we do not compare our predictions with the USLE, but rather evaluate different machine-learning models against long-term monitoring data. Our goal is to develop models capable of reproducing erosion patterns at the field-to-landscape scale, independent of the USLE framework. We thank the reviewer for highlighting this inaccuracy, and we will |
|                                                                                                                                                                                                                                                                                                                                                                                                                                                                                                                                                                                                                                                                                | revise the text accordingly to make this clearer.                                                                                                                                                                                                                                                                                                                                                                                                                                                                                                                                                                                                                                                                                                                                                                                                                                                                                                                                                                                                                                    |
| L 77: Which models?                                                                                                                                                                                                                                                                                                                                                                                                                                                                                                                                                                                                                                                            | Thank you for pointing this out. We will change phrasing to: "The study uses data from seven".                                                                                                                                                                                                                                                                                                                                                                                                                                                                                                                                                                                                                                                                                                                                                                                                                                                                                                                                                                                       |
| Data                                                                                                                                                                                                                                                                                                                                                                                                                                                                                                                                                                                                                                                                           |                                                                                                                                                                                                                                                                                                                                                                                                                                                                                                                                                                                                                                                                                                                                                                                                                                                                                                                                                                                                                                                                                      |
| Chapter Data collection: In general, this chapter does not give enough details about the sources of data, the measurement methods, their range, their resolution, and their quality. The lack of reference to the sources also makes it impossible for the reader to get an idea about these relevant aspects.                                                                                                                                                                                                                                                                                                                                                                 | We agree that the manuscript would benefit from a more thorough description of the data. We will add a detailed table to the appendix describing the predictor variables, how they were acquired and calculated, their native resolution, and other relevant information.                                                                                                                                                                                                                                                                                                                                                                                                                                                                                                                                                                                                                                                                                                                                                                                                            |
| L 95: What is the accuracy of the data? Were there independent repeated surveyors to estimate the accuracy?                                                                                                                                                                                                                                                                                                                                                                                                                                                                                                                                                                    | A version of the monitoring data collected for the years 2000 to 2016, including a description of the data collection method and accuracy assessments, is discussed in <a href="Steinhoff-Knopp & Burkhard">Steinhoff-Knopp & Burkhard</a> (2018):                                                                                                                                                                                                                                                                                                                                                                                                                                                                                                                                                                                                                                                                                                                                                                                                                           |
|                                                                                                                                                                                                                                                                                                                                                                                                                                                                                                                                                                                                                                                                                | "Our examinations (comparison of multiple measurements by different observers and data derived by structure-from-motion-methods) show an error rate of approximately 15%."  Steinhoff Knopp B, and Burkhard B. (2018): Soil presion by                                                                                                                                                                                                                                                                                                                                                                                                                                                                                                                                                                                                                                                                                                                                                                                                                                               |
|                                                                                                                                                                                                                                                                                                                                                                                                                                                                                                                                                                                                                                                                                | Steinhoff-Knopp, B. and Burkhard, B. (2018): Soil erosion by water in Northern Germany: long-term monitoring results from Lower Saxony,450 Catena, 165, 299–309, https://doi.org/10.1016/j.catena.2018.02.017                                                                                                                                                                                                                                                                                                                                                                                                                                                                                                                                                                                                                                                                                                                                                                                                                                                                        |
| How did you know there had been an erosion event, given that high-intensity rain cells have only a spatial extent of about 1 km² (see Lochbihler et al. 2017, Geophysical Research Letters)?                                                                                                                                                                                                                                                                                                                                                                                                                                                                                   | It was not always known in advance of a survey whether an erosion event had occurred. Surveys were nevertheless conducted each time (except during winter months) when a respective 1 km² grid cell recorded a rainfall event exceeding 12.7 mm (we use radar rainfall information from the German weather service) to ensure that all potential events were documented. Surveys were also conducted when farmers reported erosion features.                                                                                                                                                                                                                                                                                                                                                                                                                                                                                                                                                                                                                                         |
| L 97: What is sheet-to-linear erosion? Isn't this rill erosion, which is already in the first group?                                                                                                                                                                                                                                                                                                                                                                                                                                                                                                                                                                           | Sheet-to-linear erosion comprise erosion systems showing both features of sheet and linear erosion. In the surveys these are mapped in its own category. See also <a href="Steinhoff-Knopp & Burkhard">Steinhoff-Knopp & Burkhard</a> (2018)                                                                                                                                                                                                                                                                                                                                                                                                                                                                                                                                                                                                                                                                                                                                                                                                                                 |

L 101: Nineteen variables are pretty limited. I would not criticise this, but in L 38, you criticised a limited number of variables. Your arguments do not match. (BTW: The (R)USLE uses more than 19 variables to calculate the final six factors; hence, your data set is more limited).

Thank you for bringing to our attention that our argumentation in this regard needs to be refined and partially rephrased. Our aim is to detect patterns and relationships of soil erosion and related factors at the field-to-landscape scale. For this purpose, in addition to the factors of the USLE, we also considered other variables describing landscape characteristics that may affect erosion patterns. Many of these variables are themselves derived from multiple sub-variables (e.g. the included USLE factors). As mentioned in the discussion section of the manuscript, we acknowledge that our set of variables is not allencompassing. However, we believe they provide a sufficient starting point for our analysis.

L 110: Better call it the Pearson correlation coefficient because Pearson and even the regression have several coefficients. In the following, r is mostly in italics. Please be consistent.

Thank you for bringing this to our attention. We will fix it accordingly.

Table 1: DEM is definitely wrong because this is the entity of all elevation data. Do you mean altitude?

More details about the resolution and the quality of your DEM have to be given (see the general remark regarding the data chapter) because many of your following variables depend strongly on these two parameters.

We agree that the information represented by the respective DEM grid cells corresponds to altitude. We will clarify this in the manuscript and add more detailed information about the DEM, including its native resolution and quality, to the data table in the appendix.

How was slope length defined, in the sense of the USLE or in a geomorphological sense? Was it defined for the field or for the raster cell? I guess you did not use slope length, which would be one value for the entire slope, but you may have used the upslope length of each raster cell. I do not like guessing what you did (a similar question could be raised for almost all variables).

Flow accumulation is described as the total accumulated runoff. This would require runoff modelling because runoff will depend on soil, crops, heterogeneity of rain and other variables. I guess you mean the upslope drainage area. More explanation required!

Wetness index: What is a 'modified catchment area calculation'?

Machining direction: This will differ on different field parts because of the headland and complex topography. How was it defined? It may also vary over time.

Regarding the R and LS factors, see below. How was the C factor determined? Did you consider individual rains and the corresponding field states, or did you use some more generalised C factor? Which degree of generalisation did you use? K factor, based on which data?

The table must be complemented with statistical metrics like mean, SD, min, and max, which give an idea of the range the data covers. This is essential for the interpretation of Fig. 5.

L 178: Conventional cross-validation is inappropriate in your case because your raster cells are highly

We agree that additional details and explanations are necessary. Therefore, as mentioned above, a table providing more detailed descriptions and further information on each predictor variable, including statistical metrics, will be added to the appendix to improve the comprehensibility of our methodology. Thank you for highlighting this point.

Thank you for this comment. As part of our study, we applied a leave-one-area-out approach. We agree that these results

autocorrelated. Hence, the left-out data are not an independent data set. I suggest using a seven-fold cross-validation by leaving out one of your study areas at a time.

provide further insights, and they indeed show similar findings regarding the performance of the different models. Again, the CNN performs best. However, this approach also has its own limitations (which we will add to the discussion). Because the monitored areas differ considerably in size and characteristics, the six remaining areas in each iteration cannot fully represent the entire dataset. Since our primary goal is to compare different models and their ability to reproduce soil erosion patterns, the conventional cross-validation remains useful and will continue to be the main focus of our study. Nonetheless, the results of the seven-fold leave-one-area-out approach will be included alongside the current results and discussed accordingly as it gives insights on the transferability of the models.

L 185: I cannot see the five pairs in Table 1. Which pairs do you mean?

Thank you for bringing this to our attention. We will fix the reference accordingly.

L 187: The correlation between R and altitude is strange. I am not aware of any meteorological process that would influence rain within your altitudinal and spatial range. I guess the correlation is an artefact of an inappropriate resampling procedure. Unfortunately, resampling was not described.

The correlation is explainable: The investigation areas are situated in three different regions in Lower Saxony. Each of these regions has a typical (average) altitude correlating with typical R factors resulting from climatic conditions in Northern Germany. It is not a resampling artefact.

Fig. 4: The x-axis appears to have a log scale. Then, zero would not be possible, although shown (likely it is 0.001) and although being found in the data set. I recommend using a square-root scale, which allows for a true zero and does not compress the data in the relevant range of 0.1 to 50 t because of the inflation of the irrelevant range between 0.001 and 0.1

Thank you for this suggestion. We will revise the graph accordingly.

This also leads to the question: Were there no negative values in your data set (colluviation)? Including negative values would be a clear advantage compared to the USLE. In any case, the reason for the lack of negative values has to be explained.

Deposition sites are monitored, but due to their qualitative character, this information was not included in this study.

L 224: The high importance of altitude shows that the results of your approach lack transferability to other areas. I can easily imagine a similar erosion situation (similar topography, similar soils, similar land use, similar rain), but a few hundred meters higher (or even a few thousand meters higher if we think of a high valley in the Andes). The large importance of altitude would then cause very strange predictions. The matching of the training and the application situation is an indispensable requisite for your approach that does not restrict the input data to meaningful and universally valid variables (especially if you request unlimited variables). It is worth discussing this constraint, which is especially important in the black box of neuronal networks. Whether the variables are used meaningfully in view of the erosion process by the network is unknown and irrelevant for the result. It is, however, highly relevant for the transferability. While it is relatively easy to find out whether, for instance, the K factor equation is applicable in a specific case (e.g.,

The importance of altitude and its implication on the transferability is a fair point to discuss. However, at the landscape scale, and within the extent of our study area, a DEM can still provide valuable relative information to distinguish between different rates of soil erosion. The high importance of altitude does not indicate that absolute elevation is important, but rather that the relative altitude between grid cells is important. Nonetheless, it remains true that transferability to regions outside the training data extent can be a major limitation for all models trained on spatially restricted datasets (see chapter on limitations).

Regarding to the Andes example: The application of the models trained in Northern Germany to significant different environmental conditions without validation is not a useful application. As mentioned, the transferability of the models is limited but not tested yet. Therefore, we do not claim to create models for all agricultural settings around the world, as training data cannot reflect the needed variability. Transferability to croplands in Northern Germany can be achieved – but this is

peatland erosion), it is difficult to find out in which case a neural network result will fail when transferred to a different situation.

not tested in our study. We will add further lines to the chapter limitations to address this topic in more detail.

Fig. 5: The low importance of LS is strange, particularly because of the higher importance of flow accumulation and slope. Essentially, LS is the product of flow accumulation and slope gradient and thus must be of higher importance. Could LS be wrongly calculated by assuming straight slopes, although you have converging and diverging slopes? Furthermore, did you use the field's LS factor or the pixel's LS factor, which is entirely different information? Your M&M section requires clearly more information. Otherwise, the results cannot be understood.

At first glance, we had the same impression, as the LS factor combines several relevant pieces of topographic information. We used pixel-based LS factors (will be described in the annex), calculated using the <a href="Desmet & Govers (1996">Desmet & Govers (1996)</a> method, which includes field boundaries as implemented in SAGA GIS, and cross-checked the results.

The relatively low importance of the LS factor in our models likely results from the neural network's ability to internally reconstruct similar relationships directly from its input variables (e.g., DEM, slope, and flow accumulation). In other words, the network can already capture the relationships between slope and flow accumulation that are represented by the LS factor, making the explicitly provided LS variable partly redundant, and therefore less important. Consequently, a functional relationship between variables does not necessarily imply equal importance for the model output.

Desmet, P. J. and Govers, G. (1996): A GIS procedure for automatically calculating the USLE LS factor on topographically complex landscape units, Journal of soil and water conservation, 51, 427–433,

https://doi.org/10.1080/00224561.1996.12457102

CNN was the best method in your case. Does this have any relevance? Will CNN always or at least often be the best? We don't know because this is an unreplicated experiment. Usually, we regard unreplicated results as meaningless. I wonder whether you could improve the validity of your analysis. For instance, you could run your seven study areas separately. Is CNN the best in all seven cases? Is the ranking of variables similar in all seven cases (which would allow us to say something about transferability at least within your region)? You could run your analysis ten times with a subset of 10 randomly selected variables from your data set. Is CNN the best method in all cases? Presently, we do not know, and hence your conclusion that CNN outperforms other methods remains just a speculation.

We appreciate the reviewer's comment on the need for replication and robustness testing. To address this, we replicated our comparative analysis using a leave-one-area-out cross-validation approach, where each of the seven study areas was used once as an independent test case. The results show a consistent ranking of model performances across areas, with the CNN again achieving the highest performance metrics. While we acknowledge that this approach has its own limitations, it provides additional evidence for the robustness and relevance of our findings. The new results and discussion will be added to the manuscript accordingly.

---

## Author Comment (AC2)

**Dear Reviewer,**

We sincerely thank you for your detailed and constructive feedback. We have carefully considered each of your comments and will revise the manuscript accordingly. We are pleased that you recognize the value of our work and are confident that the revisions made in response to your suggestions will further strengthen the manuscript. Please find our detailed responses to each of your points below.

**Reviewer comments**

**1. Introduction:**

The introduction is well written and effectively prepares the reader for the paper. However, the authors largely restrict their literature review to soil erosion modelling. While this is understandable to a certain degree, the claimed novelty of the paper lies in applying "new" methods such as CNNs and multi-layer neural networks. These models, however, are not particularly novel in this context, as CNNs have been applied to soil prediction tasks at least since 2019 (e.g., Padarian et al., 2019). The study would offer stronger novelty by considering more recently proposed methods from the broader ML literature (for instance, the high-quality TabArena benchmark by Erickson et al., 2025, which compares state-of-the-art tabular learners). Several of these modern methods have already been successfully tested in soil science, and established approaches such as CatBoost have been available for even longer. I understand that it is not feasible to cover every recent method, but the current comparison does feel somewhat outdated for a paper that aims to emphasize on machine learning aspects.

Padarian, J., Minasny, B., & McBratney, A. B. (2019). Using deep learning for digital soil mapping. Soil, 5(1), 79-89.

Erickson, N., Purucker, L., Tschalzev, A., Holzmüller, D., Desai, P. M., Salinas, D., & Hutter, F. (2025). Tabarena: A living benchmark for machine learning on tabular data. arXiv preprint arXiv:2506.16791.

33: The use of the term AI does not seem appropriate in this context and comes across more as a buzzword. Since the paper exclusively discusses machine learning methods (e.g., L. 67), I suggest using machine learning consistently instead of AI.

**Response**

We thank the reviewer for the positive feedback on the introduction and for the valuable comments. We agree that CNNs have already been applied in soil science for various purposes, particularly for modelling soil properties such as soil organic carbon. However, the cited studies and similar works do not apply complex neural network architectures, such as CNNs, for quantifying continuous soil erosion rates. Erosion rates are not a soil property but a function of various natural and management factors including soil properties, erosive rainfall, topography, management, etc. The novelty of our study therefore lies in the application of complex neural networks to model patterns of continuous soil erosion rates at the field-to-landscape scale.

The focus of this study is to explore and compare neural networks with a benchmark method (Random Forest) in this context. To our knowledge, no previous study has done this and used CNNs to predict continuous soil erosion rates at this spatial scale.

Nonetheless, we acknowledge that additional, recently proposed machine-learning could also provide valuable insights and should be considered for future research and we will add these points to the discussion.

Thank you for bringing this to our attention. We agree that consistent terminology is important and will rephrase L33 accordingly.

**2. Methodology:**

I have several concerns about the hyperparameters and the validation used in this study. Other comments are of minor nature:

**Hyperparameters:**

It remains unclear how the authors tuned their models. From the description (L. 178–179), it appears that hyperparameters were adjusted directly on the validation folds of the 5-fold CV. This approach introduces data leakage, as the same data are effectively used both for model selection and for performance estimation, which reduces the penalty for overfitting. Proper hyperparameter optimisation requires a nested cross-validation scheme, where the data are split into three parts: a training set for fitting the model, an inner validation set for selecting hyperparameters, and an outer test set (or fold) for obtaining a performance estimate.

I looked into the provided code but could not find any script related to hyperparameter tuning. Instead, in the models script I found only fixed parameter settings. This is problematic, as optimal hyperparameters should be determined separately for each training fold within the cross-validation. Without such a procedure, the reported results may not reflect the best achievable model performance and risk being biased by arbitrary parameter choices.

Lastly, the search space for the hyperparameters was not given. This is extremely important for a fair model comparison, if a poorly tuned RF is compared to a well-tuned NN, the comparison would not be fair. There is a lot of studies on how this can induce bias in benchmarking (e.g., Nießl et al. 2022).

Nießl, C., Herrmann, M., Wiedemann, C., Casalicchio, G., & Boulesteix, A. L. (2022). Over-optimism in benchmark studies and the multiplicity of design and analysis options when interpreting their results. Wiley Interdisciplinary Reviews: Data Mining and Knowledge Discovery, 12(2), e1441.

Do I understand correctly that this figure shows the "ground-truth" soil erosion dataset, and that these data are available in raster format, i.e., the true (or approximate true) erosion values are known across the

We apologize for our lack of sufficient documentation regarding the hyperparameter tuning. The tuning of hyperparameters was conducted separately from the main training and validation procedure, using a grid search performed prior to the main cross-validation runs (Raschka, 2020; Yu and Zhu, 2020).

We agree that it should be described more thoroughly to avoid confusion, and we will revise the description in the manuscript accordingly. In addition, we will include the hyperparameter tuning scripts and the respective search space in the referenced repository.

Raschka S. (2020): Model Evaluation, Model Selection, and Algorithm Selection in Machine Learning, https://arxiv.org/abs/1811.12808

Yu, T. and Zhu, H. (2020): Hyper-parameter optimization: A review of algorithms and applications, arXiv preprint arXiv:2003.05689,

https://doi.org/10.48550/arXiv.2003.05689

The raster data displayed in Fig. 1 represent spatially continuous mapped erosion patterns rather than single point measurements based on erosion pins (as in Gholami et al., 2021). The dataset is based on empirical long-term

entire study area? If so, I find this somewhat questionable, since such complete "ground truth" presumably relies on interpolation or modelling itself, and may therefore not represent true independent measurements. More importantly, it is unclear why additional modelling is applied, given that each crossvalidation repetition already uses 80% of the study area for training. In digital soil mapping, modelling is typically motivated by sparse point observations, where the objective is to generate high-resolution maps from limited data. In contrast, this study seems to assume ground-truth values for every raster cell, a setup that almost inevitably leads to overly optimistic performance estimates with poor generalization value. Would a strategy such as "leave-one-validation-site-out" not provide a more realistic evaluation of model performance? I may be missing a domain-specific aspect of soil erosion mapping, but from a classical digital soil mapping perspective this design appears problematic.

For example in Gholami et al. (2021), which is also cited in this paper, they used some point data and they have specified validation points. I am missing something like this in this study. To me, this makes much more sense but I do not see this in Fig. 1.

Gholami, V., Sahour, H., & Amri, M. A. H. (2021). Soil erosion modeling using erosion pins and artificial neural networks. Catena, 196, 104902.

soil erosion monitoring data obtained in surveys, which were subsequently aggregated to a raster format to enable spatial analysis (see <a href="Steinhoff-Knopp & Burkhard">Steinhoff-Knopp & Burkhard</a>, 2018). It is not directly derived from interpolation or modelling, based on single points.

The aim of our study was to assess how well different machine-learning models can reproduce these observed erosion patterns and loss rates at the field-to-landscape scale and detect underlying relationships.

We thank the reviewer for pointing out the potential value of a "leave-one-area-out" validation approach. In fact, we applied this approach during our study, and the results also show that the CNN achieves the best predictive performance among the tested models. However, this approach also has its own limitations given the available data and was not the primary focus of our analysis. Nevertheless, we agree that it adds further validity to our results and provides insight into the models' generalizability. We will therefore include the corresponding "leave-one-area-out" results in the revised manuscript.

Steinhoff-Knopp, B. and Burkhard, B. (2018): Soil erosion by water in Northern Germany: long-term monitoring results from Lower Saxony,450 Catena, 165, 299–309, https://doi.org/10.1016/j.catena.2018.02.017

**Minor Comments**

104: I may be wrong, but the overall study areas cover a few hundred ha, but the grid of the original R-factor was 1 km x 1 km. Even if resampled (how?), is this not too broad for the study area context. Maybe a reference which refers to this procedure could be useful?

The R factor indeed shows only regional variation on a 1 km x 1 km resolution and does not differ in a relevant manner within individual study areas. But it differs between the study areas which are situated in different regions of lower Saxony resulting in different R factors. We agree that further details are needed to describe the different predictor variables and will add a table to the appendix with comprehensive information on each variable, including the R factor.

123: It would be more precise to write "a random subset of the feature [or variables]". Using a subset of data (i.e., training data) is also possible as a hyperparameter but not by definition a classical parameter in Random Forest.

Thank you for pointing this out. We agree and will change the phrasing accordingly.

| 2.3.4 It is not clear from the section but implied. Did the authors use a "2D" CNN, with what Y x Y raster cell?                                                                                                                                                                                                                     | A 2D CNN was used with 7 by 7 pixels. We will add more details to the description in the manuscript to make this clearer.                                                                                                                                                                                                                                          |
|--------------------------------------------------------------------------------------------------------------------------------------------------------------------------------------------------------------------------------------------------------------------------------------------------------------------------------------|--------------------------------------------------------------------------------------------------------------------------------------------------------------------------------------------------------------------------------------------------------------------------------------------------------------------------------------------------------------------|
| Figure 4: Why do the ECDF curves of the models appear so smooth? I would expect them, similar to the mapped erosion rate, to be step functions. This suggests that the ECDFs may have been constructed differently for the models and for the mapped erosion rate. Could the authors please clarify how these curves were generated? | The ECDFs were generated directly from all continuous erosion values by sorting the data and plotting the cumulative proportion of values ≤ x using a step function (matplotlib.pyplot.step). No interpolation or smoothing was applied. The smoother visual appearance of the model ECDFs results from the continuous and smoother nature of the model estimates. |
| Figure 5: The unit is missing. It is not simply [%], but rather increase of MSE in %. While this may be clear from the context, the figure should explicitly state the correct unit.                                                                                                                                                 | Thank you for highlighting this. The figure represents the relative permutation importance [%], which is based on each variable's normalized contribution to the total increase in MSE. We will adapt the manuscript and figure label to make this clearer.                                                                                                        |